# Polymorphism within *IGFBP* Genes Affects the Acidity, Colour, and Shear Force of Rabbit Meat

**DOI:** 10.3390/ani13233743

**Published:** 2023-12-04

**Authors:** Łukasz Migdał, Anna Migdał, Sylwia Pałka, Michał Kmiecik, Agnieszka Otwinowska-Mindur, Ewelina Semik-Gurgul, Józef Bieniek

**Affiliations:** 1Department of Genetics, Animal Breeding and Ethology, University of Agriculture in Krakow, Al. Mickiewicza 24/28, 30-059 Krakow, Poland; anna.migdal@urk.edu.pl (A.M.); sylwia.palka@urk.edu.pl (S.P.); michal.kmiecik@urk.edu.pl (M.K.); agnieszka.otwinowska@urk.edu.pl (A.O.-M.); rzbienie@cyf-kr.edu.pl (J.B.); 2Department of Animal Molecular Biology, National Research Institute of Animal Production, Krakowska 1 Street, 32-083 Balice, Poland; ewelina.semik@izoo.pl

**Keywords:** IGFBPs, rabbits, SNPs, meat quality, pH

## Abstract

**Simple Summary:**

Growth traits are important parameters in husbandry. Especially for meat-type animals, understanding the genetic background and identifying naturally occurring polymorphisms can enhance production. Also, meat quality is a very important factor for food processing and consumers. In this study, we analyse IGFBP genes for possible polymorphisms that may influence growth traits and/or meat quality traits in rabbits. We found one polymorphism that influences the pH value, one of the most important indicators of meat quality, and therefore, this polymorphism may be used in the selection process.

**Abstract:**

Rabbits are important livestock animals, popular for their nutritional value. Nowadays, the molecular background of traits influencing the quality of meat and meat products is in high demand. Therefore, in the current study, we analyse the sequences of *IGFBP1*, *IGFBP2*, *IGFBP4*, *IGFBP5*, and *IGFBP6* for possible polymorphisms. Based on a bioinformatics analysis in an association study on 466 animals of different breeds (New Zealand White × Flemish Giant crossbreed (9NZWxFG), Termond White (TW), Popielno White (PW), and Flemish Giant (FG)), we analyse the influence of five polymorphisms within the *IGFBP* genes. Statistically significant differences were found among the carcass and meat quality traits but not for all of the analysed rabbit breeds. The most promising polymorphism was g.158093018A>T within the *IGFBP5* gene. The values of pH24 of m.longissimus lumborum (m.l.l.) and biceps femoris muscles (m.b.f.) were higher for the AT genotypes compared to the AA genotypes for the TW and NZWxFG crossbreeds. Also, for pH24, we found differences in ing.41594308T>C for NZWxFG, where the TT genotype values were higher than the TC values. We found differences in L*24 on m.l.l. for g.41592248A>C for NZWxFG. For m.b.f., significant differences were found in b*45 for g.3431insAC in the FG population and a*45 for g.41592248A>C and g.158093018A>T in the TW population. The shear force statistically differed for g.158093018A>T in TW rabbits and g.41592248A>C for NZWxFG. We conclude that this polymorphism may be promising for better quality rabbit meat and may be implemented in selection processes.

## 1. Introduction

Growth is an important aspect of animal production. Growth traits are considered moderately to highly heritable. It must be said that growth is a non-linear process, and growth patterns can differ among animals, ages, and breeds [1,2]. As reported by Blasco et al. [3], growth is decisively important in the rabbit meat husbandry. Information about the genetic background of phenotypic variations and their economic importance to breeders in terms of growth, fat deposition, and meat quality should be highlighted to enhance production performance. To achieve this goal, marker-assisted selection (MAS) and genome-wide association study (GWAS) approaches should be mentioned, aiming to identify the quantitative trait loci (QTL) associated with key traits [4]. The successful implementation of genomic selection in cattle leads to increases in the annual rate of the genetic gain of low-heritability traits by 50–100% [5]. These results may be promising for other livestock animals, as identifying the genetic background of phenotypic traits could boost selection gain. The domestication process led to multi-purpose species, which include many breeds and lines with broad phenotypical traits. In addition to being used as meat species, rabbits are bred as fancy rabbits and wool-use and fur-use rabbits [6,7]. The meat of rabbits, because of its composition, especially the content of easily digestible proteins (about 20–22%), is considered a healthy food with consumer benefits. Also, its “functional” food properties qualify rabbits as sources of precious and valuable meat [8]

For rabbits, a GWAS was successfully used to identify the genetic background of the long-hair trait of Angora rabbits [7] and also for signatures of selection in fancy rabbits, like coat colour and pattern and body size [6]. There are also reports of using this method for meat-type rabbits [9,10,11,12]. However, applying genomic selection in rabbits as an evaluation method has many limitations, such as the cost of a commercial SNP platform, short generation intervals, and the low economic value of paternal rabbits [13]. Therefore, MAS [14] can be considered a reliable selection approach. In our previous experiment [15], we showed that identified SNPs may be used to improve important traits so that they may be used in selection processes, but it must be mentioned that there are differences between breeds and trait values. So far, numerous reports have identified SNPs that influence the body weight and meat parameters in rabbits [16,17,18,19], providing new information that could be used in selection processes.

A highly conserved signalling pathway involving insulin-like growth factor 1 (IGF1) plays a major role in regulating skeletal muscle growth. The activity of this pathway may be regulated by many factors and pathways at different steps. One of these factors includes IGF binding proteins (IGFBPs) with IGFB5, as it can block IGF1 by inhibiting its binding to IGF1 receptors [20]. The binding of IGF1 with IGFBPs prolongs the half-life of IGF-1 in circulation and prevents the interaction of IGF1 with insulin receptors (IRs). This possible interaction of IGF1 and IR may cause a hypoglycaemic effect since the IR and IGF1 receptors (IGF1Rs) share a similar structure [21].

Therefore, the current study analyses *IGFBP* genes for possible polymorphisms associated with the growth, slaughter, carcass, and meat quality traits in different rabbit breeds.

## 2. Materials and Methods

### 2.1. Ethical Review

The authors confirm that the experiment complied with the European Union’s Directive on Animal Experimentation (Directive 2010/63/EU) and ARRIVE guidelines. All experimental protocols were approved by the 2nd Local Institutional Animal Care and Use Committee (IACUC) in Krakow, agreement No.267/2018, and the Institutional Animal Care Review Board of the Faculty of Animal Sciences, the University of Agriculture in Krakow permissions, 29/2016, 37/2016, and 2/2018. All procedures were performed in accordance with the relevant guidelines and regulations.

### 2.2. Animals

The experiment was conducted under standardised conditions at the Experimental Station in the Department of Genetics, Animal Breeding and Ethology, University of Agriculture in Krakow. In the present study, we analysed data from 466 animals (male/female—1:1): 130 Termond White (TW) rabbits (from 20 bucks); 40 Flemish Giant (Belgian Giant Grey) rabbits (FG) (from 8 bucks); 71 Popielno White (PW) rabbits (from 10 bucks); and 225 crossbred rabbits of the F2 generation of New Zealand White × Flemish Giant (NZWxFG) (18 bucks). The crossbred population was a part of the experiment conducted at the Department of Genetics, Animal Breeding and Ethology for possibly increasing the slaughter weight of NZW rabbits by mating them with a large breed (FG). The crossbred population was derived from 48 litters, and subpopulations of the animals with the highest, medium, and lowest slaughter weights were chosen for the experiment. The purebred animals (TW, FG, and PW) were under the control of the National Centre for Animal Breeding. The animals were kept in a heated hall with a water supply (nipple drinkers), lighting (14L/10D), and exhaust ventilation. From weaning (on the 35th day of life) until the 84th day of life, the animals were kept in wire metal cages intended for the commercial rearing of rabbits (with two rabbits per cage). Water and feed were available ad libitum; their pelleted commercial diet contained 15% crude protein, 16.1% crude fibre, and 3.5% crude fat.

### 2.3. Carcass Traits

The rabbits were weaned in their fifth week of life and slaughtered in week 12 (BW12). The standardised methodology for slaughter was used as follows: After 24 h of fasting, the slaughter body weight (SW) was recorded, and the animals were subsequently slaughtered. The rabbits were stunned, immediately bled, pelted, and eviscerated. After slaughter, the hot carcass weight (HCW) was recorded, and after 24 h of storage at 4 °C, the chilled carcass weight (CCW) was recorded. The dressing-out percentages (%) were calculated—both the dressing-out percentage hot (DPH) = [HCW/SW]*100 and the dressing-out percentage chilled (DPC) = [CCW/SW]*100. Each carcass was divided between the seventh and the eighth thoracic vertebrae and between the sixth and seventh lumbar vertebrae, and the dissected meat, bone, and fat weights from each part were recorded. All measurements used Łucznik KS-205 electronic scales (Galeria Łucznik Co., Ltd., Wrocław, 121 Poland, e = 0.1).

### 2.4. Colour and pH Measurement

The colour of the meat was determined using Minolta CR-400 chroma meters (Minolta Co., Ltd., Osaka, Japan) calibrated with a white plate supplied by the manufacturer and programmed to use a D65 standard illuminant, a 10° observer with an illuminating/viewing aperture size of 11 mm, and a blooming time of 45 min. The average value of three readings was recorded for each sample and expressed as the CIE lightness (L*), redness (a*), and yellowness (b*). The pH values in the meat were determined using the Consort C561 pHmeter (calibration was performed before analysis in 4.0 and 7.0 pH buffers) with a temperature probe with temperature compensation (Consort, Turnhout, Belgium). The colouring and pH were recorded 45 min after slaughter and 24 h after chilling on *m.longissimus lumborum* (loin—m.l.l.) and *m.biceps femoris* (hind leg—m.b.f.).

### 2.5. Texture Analysis

Cylindrical samples from the *m. longissimus lumborum* were cut from the right half of the loin. The samples were vacuum-packed in foil used for food storage, frozen for 72 h at −18 °C, thawed at room temperature, and boiled in a water bath at 80 °C for 40 min. The shear force and texture parameters were measured using the TA.XTplus Texture Analyser (Stable Micro Systems Co., Ltd., Godalming, UK). The shear force was measured from three cylindrical samples (15 mm in diameter and 15 mm in height) using a Warner–Bratzler attachment and a triangular notch in the blade. Meat samples were cut perpendicular to the direction of the muscle fibres. The blade speed during the test was 2 mm/s. The texture (hardness, springiness, cohesiveness, and chewiness) was analysed using the attached cylinder, 50 mm in diameter. The three samples were subjected to a double pressing test, applying a force of 10 g to 70% of their height. The cylinder speed was 5 mm/s, and the interval between presses was 5 s.

### 2.6. Blood Collection and DNA Extraction

The DNA was collected during slaughter into tubes containing EDTA. The DNA was extracted using a WIZARD Genomic Kit (Promega, WI, USA) from 300 µL of blood collected after slaughter into tubes containing EDTA.

### 2.7. Primer Design and Sequencing

The primers for sequencing were designed with Primer3 software (v. 0.4.0) (https://bioinfo.ut.ee/primer3/) (accessed on 13 May 2020) using the sequences of rabbit *IGFBP1* (ENSOCUG00000012831), *IGFBP2* (ENSOCUG00000008561), *IGFBP4* (ENSOCUG00000002828), *IGFBP5* (ENSOCUG00000025513), and *IGFBP6* (ENSOCUG00000001412). All chromatograms were visually inspected in FinchTV v1.4.0 (Geospiza Inc., Seattle, WA, USA). The online sequence alignment BLAST tool (https://blast.ncbi.nlm.nih.gov/Blast.cgi) (accessed on 20 October 2020) was used to confirm the obtained sequences. The quality of the sequences was inspected using the CodonCode Aligner (CodonCode Corporation, Centerville, MA, USA, www.codoncode.com) (accessed on 20 October 2020). The SNPs were identified by aligning the reference sequence of OryCun 2.0 and aligned sequencing readings in MEGAX [22].

### 2.8. SNPs Analysis

The identified SNPs were analysed using PCR high-resolution meltings (HRM) and PCR- restriction fragment length polymorphism (RFLP). The primers for both methods are listed in Table 1. The PCR-HRM amplification reaction for the g.41594308T>C and g.3431insAC polymorphisms was performed in a 10 μL volume using the Quantum EVAGreen^®^HRM kit (Syngen Biotech, Wrocław, Poland) according to protocol and the MIC qPCR cycler (Bio molecular systems, Queensland, Australia) (Table 1). PCR-RFLP was performed using GoTaq G2 polymerase (Promega, WI, USA). First, 80 ng of DNA was added to the master mix and filled with nuclease-free water to a final volume of 15 ul. After visual inspection, the PCR products (10 µL) were digested using the restriction enzymes presented in Table 1. After incubation (according to protocol), the products were visualised in 4% agarose gel with a 100 bp DNA ladder (New England Biolabs, Ipswich, MA, USA) as follows: g.41592248A>C and g. 4988G>A—BccI (New England Biolabs, MA, USA), g.158093002A>T—BfaI (New England Biolabs, Ipswich, MA, USA), g.158078617C>T—BsrI (New England Biolabs, Ipswich, MA, USA), and g.158093018A>T—BfaI (New England Biolabs, Ipswich, MA, USA).

### 2.9. Statistical Analysis

Associations between the SNPs and quantitative traits within each breed were investigated using the analysis of variance using the general linear model (GLM) procedure of SAS 9.4 [23] and the following model:Yijk = µ + Gi + Sj + (G × S)ij + bNijk + eijk
where Yijk—studied traits; µ—mean of the trait; Gi—fixed effect of the i-th genotype (i = 1,2,3); Sj—fixed effect of the j-th gender (j = 1,2); (G*S)ij—interaction between genotype and gender; bNijk—linear regression of the day of slaughter; eijk—error term. The interaction between genotype and gender and the linear regression of the litter size were included in the model when significant. The Tukey test was used for multiple comparisons. The Bonferroni correction was used to account for multiple tests. The correction factor was derived from the number of SNPs tested. The significance threshold (*p* < 0.05) was divided by the number of tests. Thus, the Bonferroni-corrected significance level of 0.05/16 = 0.0031 was applied.

### 2.10. Bioinformatics Analysis

The estimation of the likelihood of SNP’s impact on proteins was analysed using PANTHER 16.0 software [24], where position-specific evolutionary preservation (PSEP) measured the length of time (in millions of years) that a position in the current protein has been preserved by tracing back to its reconstructed direct ancestors.

## 3. Results

### 3.1. SNPs Identifications

Out of the 37 identified SNPs (https://www.ebi.ac.uk/ena/data/view/PRJEB42486) (accessed on 20 January 2021), using the PCR-RFLP/PCR-HRM method, we analysed 5 SNPs. Those SNPs were located in coding regions (non-synonymous mutations), 5′ UTRs, and close to coding regions. Therefore, we hypothesised that those SNPs could have an impact on the protein function (based on bioinformatics analysis) and influence some growth, slaughter, and meat traits. In the *IGFBP1* gene sequence, we analysed mutations within intron 2_3—g.4988G>A and the insertion of AC within 3′ UTR. For the *IGFBP4* gene, a missense mutation in exon 3 g.41592248A>C resulted in an I182L substitution (Pdel = 0.57 with a preservation time of 456′ was probably damaging), and g.41594308T>C on the last position of 5′ UTR. For the *IGFBP5* gene, a substitution g.158093018G>T in 5′ UTR was analysed. The analysed polymorphism frequencies are presented in Table 2. Due to the heterozygosity in FG rabbit populations and almost no minor allele homozygotes for g.4988G>A, we excluded those SNPs from the statistical analysis.

### 3.2. Association Analysis

The association results with the growth and slaughter traits are presented in Appendix A, and those with carcass and meat traits are in Appendix A for g.3431insAC SNP (*IGFBP1*). Appendix A show the results for g.41594308T>C SNP (*IGFBP4*), Appendix A show those for g.41592248A>C SNP (*IGFBP4*), and Appendix A show them for g.158093018A>T (*IGFBP5*). The statistical analysis did not show significant differences between the genders, so in the tables, those results are excluded. Moreover, the interaction between gender and genotypes was not significant. Table 3 presents the significant associations between the SNPs and growth and slaughter traits. For the TW rabbits, significant differences were found between the CC genotypes and Cins for the IB weight for g.3431insAC. For the HCW in the PW rabbits, the TT genotypes were significantly higher than the TC genotypes for g.41594308T>C. We did not find statistical differences in the slaughter weight or cut weight for any of the analysed breeds. Statistical analysis did not show significant differences between the genders, so those results are excluded from the tables. Moreover, the interaction between gender and genotypes was not significant.

Table 4 presents the significant associations between the SNPs, the carcass, and the meat traits. For the FG rabbits, in m.b.f., the b45* value showed significant differences between the GG and G/GAC genotypes for g.3431insAC. The pH24 values measured on m.l.l. for the TT genotypes were significantly higher than the TC genotypes for g.41594308T>C in the NZWxFG population. For g.41592248A>C within *IGFBP4*, the AA genotypes had significantly higher a* values after 45 min on m.b.f. compared to the AC for the TW rabbits. For the NZWxFG rabbits, the L* value on m.l.l. after 24 h and the shear force for the AA genotypes showed significantly higher values than the AC genotypes. The most promising results were obtained for g.158093018A>T within *IGFBP5*. For the carcass traits, the pH24 values measured on m.l.l. and m.b.f. showed significantly higher values for the AT genotypes compared to the AA genotypes for the TW and NZWxFG rabbits. Also, for the TW rabbits, a *45 value measured on m.b.f. showed a significantly higher value for AT compared to AA. We did not find significant differences in the a* and b* values after 24 h of chilling on both muscles. The shear force of the TW rabbits was lower for the AT genotypes compared to AA for g.158093018A>T. Meanwhile, for the NZWxFG rabbits, the AC genotypes had a significantly lower shear force value than AC for g.41592248A>C. We did not find differences in the texture profile analysis values. The statistical analysis did not show significant differences between the genders, so those results are excluded from the tables. Moreover, the interaction between gender and genotypes was not significant.

## 4. Discussion

Our results highlight the influence of the g.158093018A>T polymorphism within *IGFBP5* on a meat quality parameter—pH24—as the main point of discussion. Meat quality is an important parameter that modern husbandry pays attention to. The quality of meat depends on many traits, which are affected by different genes and, therefore, different metabolic pathways. As reviewed by Blasco et al. [3], no single gene is reported that would affect the growth of rabbit meat quality. Also, feeding slaughter rabbits constitutes up to 45.3% of all costs in industrial rabbit production (animals slaughtered at 63 days of age), and therefore, it seems reasonable to analyse the candidate genes from the growth pathway for their influence on growth and carcass traits [3]. During the last few years, there has been a significant increase in reports regarding molecular approaches to rabbits’ economically important traits. Recently, there have been reports on using high-throughput methods in rabbit populations [9,10,11,12,25]. Yang et al. [12] used a genome-wide association study (GWAS) to identify traits associated with molecular markers for growth, carcass, and meat quality and associations with SNPs were shown within the *FGD4* and *DNM1L* genes. Sanchez et al. [9] investigated the molecular background of the average daily gain (ADG) associated with candidate genes located on chromosomes in the 3, 5, 6, 9, 12, 13, 16, 17, and 21 regions. None of our investigated genes were located within the regions mentioned above. The reports mentioned above used different rabbit breeds compared to those used in our experiment. This may explain the differences, as many different rabbit breeds are selected based on different traits.

As presented in the tables, many traits that impact the quality of carcasses were associated with the analysed SNPs; however, these results are limited to one or two breeds. In the present study, we analysed phenotypically different breeds: a common broiler breed—Termond White (TW), and a crossbreed of another broiler breed—New Zealand White and Flemish Giant (NZWxFG). Also, we included in the analysis a native breed (Popielno White rabbits—PW), a Polish medium breed with good maternal and growth traits [26], and the Flemish Giant, a well-known large rabbit breed. Our analysis found the most interesting results for g.158093018A>T polymorphism. For the TW and NZWxFG populations, we found that this polymorphism may be a good indicator of a very important meat quality trait—the pH value after 24 h.

Meat quality is a composite concept and is difficult to measure simply [27]. From many studies performed, it is clear that the pH is the most important indicator of meat quality, as it is responsible for the eating attributes of meat and, very importantly, its suitability for processing. Also, the pH is a major determinant of beef and pork tenderness [28]. We are aware that comparing these results with reports for other animals, like beef and pork, due to differences in the histology and processes of these meats, may be misleading, but we want to point out the importance of this parameter. In the concept of marker-assisted selection (MAS), functional DNA markers are generated, and so far, in rabbit breeding, there are plentiful markers that may be used for selection [2]. Here, we report another marker that may be useful in selecting rabbits.

The pH of rabbit meat is well described in the literature. As there are many rabbit breeds, different breeding systems that affect the pH value and differences between muscles were observed. Additionally, pH was reported to have a reliable estimate of heritability [3], and therefore, this trait may be used in selection. Paci et al. [29] did not find differences in the pH24 between local Italian and hybrid rabbits. In contrast, Chodova et al. [30] found significant differences in the pH affected by genotype, with a higher value for hybrid rabbits. Hulot and Ouhayoun [31] showed that the ultimate pH is higher in m.b.f. as a result of its lower glycolytic potential and differs between the rabbit lines selected for different traits. In our experiment, the pH of m.b.f. was higher than for m.l.l. after 24 h of chilling, which is in agreement with other reports, as it corresponds to a more oxidisable muscle [32]. In particular, the high value of pH for TW rabbits on m.b.f. (6.11 ± 0.06), according to the results from other species, it may indicate dark, firm, dry (DFD) meat, which is not acceptable to consumers. Kowalska et al. [33] assigned meat from rabbits with a pH24 of 5.22 ± 0.92, measured on m. longissimus as PSE; however, this was based on what is commonly used for poultry and pork. These results may be temporary, as there is a shortage of information about this meat defect in rabbits, and very often, a comparison is made to reference values used in pork and poultry meat. However, the meat from NZWxFG seems to be within an acceptable pH range. Rejduch et al. [34] found differences in the expression of the *IGFBP5* mRNA. In particular, these authors found under-expression in the Duroc breed. Acidity affects meat processing suitability and its culinary uses. Defects such as pale, soft, exudative (PSE) and dark, firm, and dry (DFD) lead to a lower culinary value of meat and decreased consumer acceptance. In this context, the halothane gene can be mentioned as a good example in pigs [35]. The slandered range of pH ensures the highest technological quality for processing. Guerrero et al. [36] analysed meat from pigs with pH24 > 6.2 and pH24 < 5.8 and reported that products from meat with a higher pH24 were softer. Therefore, the slicing process may be difficult, and finally, consumer acceptance of the final product may be diminished. In Table 4, the influence on shear force is also reported—the AA genotypes for TW had significantly higher values. This may be the result of phenotypic differences among breeds and needs to be further investigated.

There are no reports about the associations of SNPs within rabbits between the *IGFBP5* and pH values. For pigs, on the SSC15 region, SNPs between 114.8 and 121.4 MB affect, among other factors, the level of residual glycogen and glucose, pH36, drip loss, and the colour parameters L*, a*, and b* were also identified [37,38]. This region harbours the most influential gene—*PRKAG3* (at 120.86 MB), close to the *IGFBP5* gene (at 118.87 MB). In rabbit chromosome 7 (OCU7), where the *IGFBP5* is mapped (at 158.1 MB), the *PRKAG3* gene is also located (at 160 MB), which may suggest that this region is important for meat traits in rabbits. As mentioned above, most traits are polygenic, which can also be seen in the pH, as reports show the influence of different polymorphisms within genes on the pH. Wang et al. [39] showed that SNPs at position 536 bp in intron 5 (C>T) of *POU1F1* influence the pH of m. biceps femoris. Also, Wang et al. [40] showed that SNPs within *CAST* (g.16443397 T>G) influence the pH of both muscles.

Colour is considered one of the most important characteristics of meat quality that most influence consumer choice [30]. Meat colour is affected by the myoglobin level and differs mostly within fibre types [30]. For the L* value, Paci et al. [29] showed significantly higher values for Hyplus rabbits compared to a local breed. In the experiment of Chodova et al. [30], Hyplus rabbits had average L* values compared to Czech local rabbits. For the b* parameter on m.b.f., Wang et al. [40] found in the *CAST* gene that the GT (2.46 ± 0.25) genotype was significantly higher compared to GG (1.81 ± 0.23) and TT (1.07 ± 0.19) in Hyla, Champagne, and Tianfu black rabbits. Our results for each analysed breed were lower for the b* value on m.b.f. compared to those reported by other authors [22,23,33]. These differences may be due to the housing systems and breeds used, which were mentioned by the authors stated above [29,30]. From the consumer’s point of view, decisions about meat quality are constantly changing. Previously, choices were made mostly based on the price of meat, income, and availability. More recently, consumers have made choices based on healthiness and sensory quality [41] and, most recently, on the sustainability of food [42]. As Maj et al. [43] reported, rabbit meat is considered one of the lightest among meat production animals, and the colour intensity is very low. In our study, we obtained similar results for the L* value after 24 h for NZWxFG (Table 4) compared with the studies by Maj et al. [43], Chodova et al. [30], and Wang et al. [44]. For the a* and b* values, significant differences (Table 4) were found for m.b.f., but only 45 min after slaughter. Consumers are not able to distinguish small differences between yellowness [44], but they assess the colour of meat after chilling, which may limit our results obtained for the a* and b* values. Altmann et al. [45] reported that consumer preferences for meat colour differ globally and can depend on demographic factors.

Rabbit meat texture may differ among breeds, thermal treatments, the composition of muscle fibres, and the transport and handling before slaughter [46,47,48,49]. For both polymorphisms—g.158093018A>T and g.41592248A>C—heterozygotes had significantly lower shear force values than homozygotes. Lower shear force values may be favourable to consumers. The results obtained among breeds differ, which is in agreement with the results of Pla et al. [32] for the shear force rather than the weight of animals.

## 5. Conclusions

Here, we have analysed the sequences of rabbit genes from the *IGFBP* family for possible SNPs and their association with growth and meat traits. We found that g.158093018A>T in Termond White rabbits and Flemish Giant and New Zealand White crossbreed rabbit populations influenced the pH after 24 h on m. longissimus lumborum and m. biceps femoris. Also, other associations were found, like shear force and colour parameters, but these findings were breed-specific. As in rabbit husbandry, a number of different breeds that are used in different countries and companies may be mainly limited to molecular approaches for selection. However, these findings may be interesting, as the pH value is one of the most important indicators of meat quality. There is a lack of reports about meat conditions for rabbits, like what exists for pigs and poultry (PSE). We hypothesised that our results might be interesting in future research to help with a better understanding of unacceptable meat conditions in rabbits. Based on our results, we hypothesise that primitive breed populations (like PW and FG) do not carry this mutation, and this SNP may occur in meat-type breeds (in our research, the TW and NZW component), which may encourage additional analyses regarding the influence of breeds on meat quality. As additional analyses must be performed on other meat and local rabbit breeds, this SNP may be used in the future as a marker in MAS.

## Figures and Tables

**Table 1 animals-13-03743-t001:** Primers designed for PCR-RFLP and PCR-HRM analysis.

Primer Pair (Gene)	Forward and Reverse (5′->3′)	bp	T_a_	Method
g.4988G>A (*IGFBP1*)	F: GAAAAGCGGGTTCAGAGAGG R: AAGGGCAAGAGACAGTGAGC	150	65	PCR-RFLP for BccI G-150 bp, A-101 + 49 bp
g.3431insAC (*IGFBP1*)	F: CAAATGCCACCAGCATTTTA R: TGTGTTCTGAGGATAAATACACCA	97	60	PCR-HRM
g.41592248A>C (*IGFBP4*)	F: GTTCCTGCCAGAGTGAGCTG R: CTGCTTGGGGTGGAAGTTG	128	65	PCR-RFLP for BccI A-93 + 35 bp, C-128 bp
g.41594308T>C (*IGFBP4*)	F:TCTGAATTCATTCCTCTATCTACCC R: GGTCAATACATGTTTTCAGATGG	58	62	PCR-HRM
g.158093018A>T (*IGFBP5*)	F: GATTGGTCGGGGAGAGAAAG R: CTTTTCGGAGGAATGGAATG	148	60	PCR-RFLP for BfaI A-148 bp, T-117 + 31 bp

bp—product length (bp); Ta—annealing temperature. As follows: g.41592248A>C and g. 4988G>A– BccI (New England Biolabs, Ipswich, MA, USA), g.158093002A>T—BfaI (New England Biolabs, Ipswich, MA, USA), g.158078617C>T—BsrI (New England Biolabs, Ipswich, MA, USA), and g.158093018A>T—BfaI (New England Biolabs, Ipswich, MA, USA).

**Table 2 animals-13-03743-t002:** Allele and genotype frequency of analysed polymorphisms.

Polymorphism	Breed	Allele		Genotypes			*p*-Value ^4^	MAF ^5^	H ^6^	He ^7^	PIC ^8^
g.4988G>A (*IGFBP1*)		G	A	GG	GA	AA					
	TW ^1^	78.46 ^2^	21.54	59.23 (77) ^3^	38.46 (50)	2.31 (3)	0.11	0.21	0.5	0.34	0.39
	FG	50	50	-	100 (40)	-	-				
	PW	92.25	7.75	84.51 (60)	15.49 (11)	-	0.47	0.07	0.26	0.14	0.23
	NZWxFG	93.78	6.22	87.56 (197)	12.44 (28)	-	0.33	0.06	0.22	0.12	0.19
g.3431insAC (*IGFBP1*)		G	GAC	GG	G/GAC	GAC/GAC					
	TW	59.62	40.38	36.15 (47)	46.92 (61)	16.92 (22)	0.77	0.4	0.62	0.48	0.54
	FG	82.5	17.5	72.5 (29)	20 (8)	7.5(3)	0.051	0.17	0.43	0.29	0.38
	PW	93.66	6.34	87.32 (62)	12.68 (9)	-	0.56	0.06	0.22	0.12	0.2
	NZWxFG	72	28	49.78 (112)	44.44 (100)	5.78 (13)	0.12	0.28	0.55	0.4	0.45
g.41594308T>C (*IGFBP4*)		T	C	TT	TC	CC					
	TW	71,15	28,85	48.46 (63)	45.38 (59)	6.15 (8)	0,22	0.28	0.56	0.41	0.46
	FG	77.5	22.5	55.0 (22)	45.0 (18)	-	0.06	0.22	0.5	0,35	0.37
	PW	63.4	36.6	45.07 (32)	36.62 (26)	18.31 (13)	0.21	0.36	0.63	0.46	0.55
	NZWxFG	60.89	39.11	40.89 (92)	40 (90)	19.11 (43)	0.06	0.39	0.55	0.48	0.45
g.41592248A>C (*IGFBP4*)		A	C	AA	AC	CC					
	TW	72.4	27.6	47.2 (59)	50.4 (63)	2.4 (3)	0.96	0.27	0.52	0.4	0.41
	FG	100	-	100 (40)	-	-					
	PW	97.89	2.11	97.89 (68)	4.23 (3)	-	0.85	0.02	0.08	0.04	0.08
	NZWxFG	93.78	6.22	87.56 (197)	12.44 (28)	-	0.33	0.06	0.22	0.12	0.19
g.158093018A>T (*IGFBP5*)		A	T	AA	AT	TT					
	TW	86.54	13.46	73.08(95)	26.92(35)	-	0.076	0.13	0.39	0.23	0.32
	FG	100	-	40	-	-	-				
	PW	100	-	71	-	-	-				
	NZWxFG	89.56	10.44	79.11(178)	20.99(47)	-	0.08	0.1	0.33	0.19	0.28

^1^ TW—Termond White rabbits; FG—Flamish Giant rabbits; PW—Popielno White rabbits; NZWxFG–crossbreeds of New Zealand White and Flamish Giant rabbits. ^2^ Allele frequencies. ^3^ Numbers of observations. ^4^ Hardy–Weinberg equilibrium *p* > 0.05. ^5^ Minor allele frequency. ^6^ Heterozygosity. ^7^ Expected heterozygosity. ^8^ Polymorphic informative content (PIC).

**Table 3 animals-13-03743-t003:** Associations between polymorphisms within *IGFBP*s genes and growth and slaughter traits.

Traits ^3^	TW ^1^					NZWxFG					PW					FG				
	Means	SD ^2^	Means	SD	*p*-Value	Means	SD	Means	SD	*p*-Value	Means	SD	Means	SD	*p*-Value	Means	SD	Means	SD	*p*-Value
g.3431insAC																				
	GG		G/GAC			GG		G/GAC			GG		G/GAC			GG		G/GAC		
IB	43 ^a^	2	36 ^b^	1	0.0025	42	1	42	1	*0.5479*	35	1	42	1	0.2810	45	2	50	4	0.4808
g.41594308T>C																				
	TT		TC			TT		TC			TT		TC			TT		TC		
HCW	1498	39	1456	24	0.6225	1420	30	1357	28	0.2894	1512 ^a^	26	1322 ^b^	33	0.0028	1657	79	1791	86	0.3085

^1^ TW—Termond White; NZWxFG—crossbreeds of New Zealand White and Flemish Giant; PW—Popielno White; FG—Flemish Giant. ^2^ SD—standard deviation. ^3^ HCW—hot carcass weight (g); IB—bones in intermediate part (g). ^a,b^ Values within a row and breeds with different superscripts differ significantly at *p* < 0.003.

**Table 4 animals-13-03743-t004:** Associations between polymorphisms within *IGFBP*s genes and carcass traits.

Traits	TW ^1^					NZWxFG					PW					FG				
	Means	SD ^2^	Means	SD	*p*-Value	Means	SD	Means	SD	*p*-Value	Means	SD	Means	SD	*p*-Value	Means	SD	Means	SD	*p*-Value
g.3431insAC																				
	GG		G/GAC			GG		G/GAC			GG		G/GAC			GG		G/GAC		
b*_45bf_	1.01	0.29	1.10	0.29	0.0887	0.99	0.14	1.04	0.14	0.2184	0.45	0.24	0.57	0.43	0.9651	1.14 ^a^	0.23	−0.50 ^b^	0.63	0.0001
g.41594308T>C																				
	TT		TC			TT		TC			TT		TC			TT		TC		
pH_24 ll_	5.75	0.03	5.83	0.04	0.5787	5.66 ^a^	0.02	5.60 ^b^	0.02	0.0011	5.84	0.04	5.82	0.05	0.1901	5.98	0.07	5.89	0.06	0.7830
g.41592248A>C																				
	AA		AC			AA		AC			AA		AC			AA		AC		
a*_45bf_	4.28 ^a^	0.26	2.93 ^b^	0.13	0.0029	11.24	0.19	11.25	0.12	0.8620	3.05	0.15	2.25	0.27	0.6495	3.55	0.22	-	-	*-*
L*_24 ll_	54.51	0.48	56.38	0.34	0.2106	57.26 ^a^	0.31	56.16 ^b^	0.21	0.003	56.55	0.30	55.06	0.88	0.0237	59.12	0.58	-	-	*-*
SF _ll_	1.90	0.11	2.01	0.09	0.0167	4.26	0.15	3.65	0.10	0.0026	1.73	0.07	1.89	0.16	0.4846	2.13	0.08	-	-	-
g.158093018A>T																				
	AA		AT			AA		AT			AA		AT			AA		AT		
a*_45 bf_	2.97 ^a^	0.17	4.14 ^b^	0.24	0.0010	11.74	0.29	11.09	0.11	0.0654	2.94	0.14	- ^3^	-	*-*	3.55	0.22	-	-	-
pH_24 bf_	5.80 ^a^	0.02	6.11 ^b^	0.06	<0.0001	5.64 ^a^	0.03	5.77 ^b^	0.02	0.003	5.97	0.03	-	-	*-*	6.08	0.02	-	-	-
pH_24 ll_	5.72 ^a^	0.03	5.91 ^b^	0.03	0.0025	5.52 ^a^	0.04	5.62 ^b^	0.01	0.0011	5.81	0.03	-	-	*-*	6.01	0.04	-	-	-
SF_ll_	2.25 ^a^	0.10	1.74 ^b^	0.08	0.0005	3.30	0.21	3.95	0.10	0.0192	1.77	0.07	-	-	*-*	2.13	0.08	-	-	-

^1^ TW—Termond White; NZWxFG—crossbreeds of New Zealand White and Flemish Giant; PW—Popielno White; FG—Flemish Giant. SF—Shear force; ll—m.longissimus lumborum; bf—m. biceps femoris. ^2^ SD—standard deviation. ^3^ In our analysis, we did not find marked genotypes in the breed. ^a,b^ Values within a row and breeds with different superscripts differ significantly at *p* < 0.003.

## Data Availability

Information about identified polymorphisms and their frequency were deposited in the European Nucleotide Archive (EVA)—https://www.ebi.ac.uk/ena/data/view/PRJEB42486 (accessed on 20 January 2021).

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
