# Peer review of "Polymorphism within IGFBP Genes Affects the Acidity, Colour, and Shear Force of Rabbit Meat"

_animals, 2023, doi:10.3390/ani13233743_

Round 1

Reviewer 1 Report

Comments and Suggestions for Authors

This paper should be very interesting for selecting rabbits for meat production. Moreover, I think that the authors must reduce the material included in the text,with particolar attention to the Supplementary Tables, full of data and difficult to read for the lack of specifications. For example:

- Table S1 and S3: most of the abbreviations in Table are not reported in the note (form IM to the end)

- Table S5, S7: in the note are present FP, IP and HP, not reported in table (???)

- Table S2, S4, S6, S8: I suggest to keep off pH, L*, a*, b* measured at 45min because consumers is not intersted in this values.

Regarding the text and the Title, I suggest to put the entire world and not the abbreviations for a better comprehension.

Table 2, 3 and 4: put every Table in one sheet. they are not easy to understand in this way.

Comments on the Quality of English Language

revise the English, the space and dots!! a lot of little errors!

Author Response

Many thanks for the rapid review and meaningful comments to the submitted manuscript. We are grateful for the time that you have devoted and have highlighted the changes within the body of the text. We believe that the changes made will improve the readership of our submission. If there are additional questions or concerns of clarity, please do not hesitate to contact our team immediately. As it was suggested we performed English editing

This paper should be very interesting for selecting rabbits for meat production. Moreover, I think that the authors must reduce the material included in the text, with particolar attention to the Supplementary Tables, full of data and difficult to read for the lack of specifications. For example:

- Table S1 and S3: most of the abbreviations in Table are not reported in the note (form IM to the end)

It was corrected – In those tables we added explanation of -IM - meat in intermediate part (g); IB - bones in intermediate part (g); IF - dissectible fat in intermediate part (g); HM - meat in hind part (g); HB - bones in hind part (g); HF - dissectible fat in hind part(g)

- Table S5, S7: in the note are present FP, IP and HP, not reported in table (???)

It was corrected and removed

- Table S2, S4, S6, S8: I suggest to keep off pH, L*, a*, b* measured at 45min because consumers is not intersted in this values.

yes that’s true, but if reviewer agree would like to keep it here to show differences between those values (pH + color 45 and pH + color 24) to show that our results are similar to obtained by other authors

Regarding the text and the Title, I suggest to put the entire world and not the abbreviations for a better comprehension.

it was done

Table 2, 3 and 4: put every Table in one sheet. they are not easy to understand in this way.

It was corrected and tables are now in one page for better and easier analyse of their content

Reviewer 2 Report

Comments and Suggestions for Authors

This was an interesting study conducted to analyse sequences of rabbit genes from the IGFBPs family for possible SNPs, and their association with growth and meat traits. The authors reported a potential SNP that significantly influences rabbit meat pH.

As mentioned by the authors, there are fewer documented reports on screening SNPs for IGFBP genes in rabbits. Hence this study may be of interest to many.

Below mentioned are some of my suggestions:

The conclusion is too short and needs to be improvised. The authors may also mention the research gaps and future perspectives of this study.

Line 313: It may be too quick to make a statement like “Also this SNP may be used as significant marker in MAS”. It can be proposed to be a potential marker for meat trait in rabbits but a recommendation for MAS can be made provided it has been proved to play a significant role to influence this trait (pH) when compared to other relevant genes.

Did the authors perform the bioinformatic analysis for the identified SNPs, like for example, if it was synonymous or non-synonymous?

Minor suggestions:

1.     Line 41: Kindly elaborate the term GWAS as it was used for the first time here in the manuscript

2.     Line 46: Kindly elaborate the term MAS

3.     There are some minor language edits to be made throughout the manuscript. For instance, in line 57 “… traits in different rabbits breed it would be appropriate to state is as “…  traits in different rabbit breeds

4.     Table 1: the word ‘rewers’ seems to be typed incorrectly, kindly rectify it. Likewise the symbol used to indicate the 3’ end was also incorrect (3”)

5.     Line 142: the content indicated here begins abruptly and seems like there’s a missing link. Kindly look into it

6.     The manuscript needs to looked into for language and/or grammar checks and also spacing errors.

Comments on the Quality of English Language

Like mentioned earlier, the manuscript had some grammatical errors which could be looked into. For example, the sentence between lines 278-279 is difficult to understand and there seems to be a language issue with it.

Likewise, there's issue with spacing between words and sentences at certain points in the manuscript.

Author Response

Many thanks for the rapid review and meaningful comments to the submitted manuscript. We are grateful for the time that you have devoted and have highlighted the changes within the body of the text. We believe that the changes made will improve the readership of our submission. If there are additional questions or concerns of clarity, please do not hesitate to contact our team immediately. As it was suggested we performed English editing

This was an interesting study conducted to analyse sequences of rabbit genes from the IGFBPs family for possible SNPs, and their association with growth and meat traits. The authors reported a potential SNP that significantly influences rabbit meat pH.

As mentioned by the authors, there are fewer documented reports on screening SNPs for IGFBP genes in rabbits. Hence this study may be of interest to many.

Below mentioned are some of my suggestions:

The conclusion is too short and needs to be improvised. The authors may also mention the research gaps and future perspectives of this study.

We rewrite conclusion:

……………. Also other associations were found like shear force and color parameters but those finding were breed specific. As in rabbit husbandry number of different breeds that are used in different countries and companies may be main limitation for molecular ap-proaches in selection. However, these findings may be interesting as pH value is one of the most important indicators of meat quality. There is lack of reports about meat conditions for rabbits like for pigs or poultry (PSE). We hypothesised that our results may be interest-ing in future research in help with better understanding not acceptable meat conditions in rabbits. Based on our results we hypothesised that primitive breeds population (like PW and FG) did not carry this mutation and this SNP may occur in meat-type breeds (in our research TW and NZW component) which may encourage to additional analysis about influence of breeds on meat quality. As there must be performed additional analysis on other meat or local rabbits breeds this SNP may be used in future as marker in MAS.As there must be performed additional analysis on other meat or local rabbits breeds this SNP may be used as marker in MAS.

Line 313: It may be too quick to make a statement like “Also this SNP may be used as significant marker in MAS”. It can be proposed to be a potential marker for meat trait in rabbits but a recommendation for MAS can be made provided it has been proved to play a significant role to influence this trait (pH) when compared to other relevant genes.

We removed world *significant*. Indeed it was too quick and probably bold from me.

Did the authors perform the bioinformatic analysis for the identified SNPs, like for example, if it was synonymous or non-synonymous?

Yes – every SNP within coding region was analysed if it is synonymous or non-synonymous mutation and if mutation was non-synonymous we performed additional bioinformatic analysis (like Panther) We added in 3.1 SNPs identifications in line 3 “Those SNPs were located in coding regions (non-synonymous mutations), 5’ UTRs, and close to coding regions, therefore we hypothesised those SNPs could have impact on protein function(also based on bioinformatics analysis)and influence on some growth, slaughter and meat traits”

Minor suggestions:

  1. Line 41: Kindly elaborate the term GWAS as it was used for the first time here in the manuscript – it was explained in line 35 but if reviewer suggest we can put there full
  2. Line 46: Kindly elaborate the term MAS – it was explained in line 36
  3. There are some minor language edits to be made throughout the manuscript. For instance, in line 57 “…traits in different rabbits breed” it would be appropriate to state is as “…  traits in different rabbit breeds” – language check were made and those mistakes were corected
  4. Table 1: the word ‘rewers’ seems to be typed incorrectly, kindly rectify it. Likewise the symbol used to indicate the 3’ end was also incorrect (3”) – it was corrected
  5. Line 142: the content indicated here begins abruptly and seems like there’s a missing link. Kindly look into it -  we changed it
  6. The manuscript needs to looked into for language and/or grammar checks and also spacing errors. - language check were made and those mistakes were corrected

Reviewer 3 Report

Comments and Suggestions for Authors

In the present paper several elements are weak points:

a) the title - significant effect on pH24 was found only for NZWxFG, and in the tables are presented also other significances and results, so it does not seem adequate to have title including only pH; title does not corresponds to the aim

b) Introduction seems to be too general and more specific information regarding previous similar research on rabbit meat quality would be beneficial

c) four tables presented in the paper must be better organised and presented; 

d) suitability of presented statistical model is questionable for obtained results i.e. it is not clear whether obtained significant differences are result of polymorphism itself and/or other effects (sex, breed, premortal procedures and/or their interactions)....

e) discussion and comparison with beef and pig meat seems inadequate due to different histology and processes in these meats compared to rabbits

f) conclusion is only focused on pH (again, some concerns arise regarding presented model); if 12 Tables are present, it seems that some other conlcusion could be made from them 

Comments on the Quality of English Language

English proofreading and editing would be beneficial. 

Author Response

Many thanks for the rapid review and meaningful comments to the submitted manuscript. We are grateful for the time that you have devoted and have highlighted the changes within the body of the text. We believe that the changes made will improve the readership of our submission. If there are additional questions or concerns of clarity, please do not hesitate to contact our team immediately. As it was suggested we performed English editing

In the present paper several elements are weak points:

  1. a) the title - significant effect on pH24 was found only for NZWxFG, and in the tables are presented also other significances and results, so it does not seem adequate to have title including only pH; title does not corresponds to the aim

Significant effect was found on pH24 also for Termond White (TW) while in PW and FG (primitive breeds) we did not identified this SNP that’s why we focused mostly in title on this parameter. Thank you for this suggestion as of course our title not correspond to all identified significant effects even if they are breed specific. We suggest title “Polymorphism within  IGFBPs genes influence on the acidity, colour and shear force of rabbit meat”

  1. b) Introduction seems to be too general and more specific information regarding previous similar research on rabbit meat quality would be beneficial

we rewrite introduction and add information about researches conducted on rabbit meat quality

  1. c) four tables presented in the paper must be better organised and presented

we reorganized tables – they are on one page and we  divide breeds and we hope it will make easier to understand our results

  1. d) suitability of presented statistical model is questionable for obtained results i.e. it is not clear whether obtained significant differences are result of polymorphism itself and/or other effects (sex, breed, premortal procedures and/or their interactions)....

at the end of 3.2 Association analysis after each table description we added “Statistical analysis did not showed significant differences between gender so in the tables those results were excluded. Moreover interaction between gender and genotypes was not significant.”

Associations between SNPs and quantitative traits within each breeds were investigated using analysis of variance through the general linear model (GLM) procedure of SAS 9.4 [16] and the following model:  word “each “ were added to show that we performed analysis for each analysed breed

In 2.2 we added :The rabbits were weaned in the 5th week of life and slaughtered in week 12 (BW12).  Standardized methodology for slaughter were as fallowed: After 24h fasting, the slaughter body weight (SW) was recorded, and animals were subsequently slaughtered

  1. e) discussion and comparison with beef and pig meat seems inadequate due to different histology and processes in these meats compared to rabbits

We are aware of this however due to lack of any information about SNPs within IGFBPs genes in rabbits we used literature available for pigs for a comparisons of obtained results. We added comments in paragraph “We are aware that comparing those results with reports for other animals like beef and pork due to differences in histology and processes of this meats may be misleading but we wanted to point out importance of this parameter”

  1. f) conclusion is only focused on pH (again, some concerns arise regarding presented model); if 12 Tables are present, it seems that some other conlcusion could be made from them

we rewrite conclusion and add comments to other traits

Reviewer 4 Report

Comments and Suggestions for Authors

The manuscript entitled “Polymorphism within one of IGFBPs genes influence on pH values after 24h storage in different rabbit (Oryctolagus cuniculus) breeds” is important in terms of providing valuable information. I have made some recommendations for improving the proposed paper.

1. The manuscript contains some syntax errors and misspellings. Revise the text to improve readability. Linguistic revision is strongly suggested. 

2. When the article is examined, the aim of the article is stated by the authors as "Therefore, the current study analyze IGFBPs genes for possible polymorphisms associated with growth, slaughter, carcass and meat quality traits in different rabbits breed". The article title is "Polymorphism within one of IGFBPs genes influence on pH values after 24h storage in different rabbit (Oryctolagus cuniculus) breeds". This title is not suitable for the content and purpose of the article. The article title should be rearranged in accordance with the content and purpose of the article.

3. When Table 2 is examined, two different expressions (“-“ or “0”) are used to express the absence of some alleles and genotypes. One of these may be preferred here.

4. The conclusion section is written only for pH 24 and is quite insufficient in terms of comprehensiveness. However, in the purpose and content of the article, growth, slaughter, carcass, and meat quality characteristics were also examined. Therefore, the result needs to be expanded and rearranged to include all examined parameters.

5. Abbreviations had been made, but these abbreviations were not explained in the article (for example Cins).

6. The article provides very valuable information, but there are some problems in the presentation of this information. Especially the tables are given very confusingly. Understanding these tables is quite problematic. For this reason, redesigning the tables in a more understandable way will add value to the article.

Comments on the Quality of English Language

The manuscript contains some syntax errors and misspellings. Revise the text to improve readability. Linguistic revision is strongly suggested. 

Author Response

Many thanks for the rapid review and meaningful comments to the submitted manuscript. We are grateful for the time that you have devoted and have highlighted the changes within the body of the text. We believe that the changes made will improve the readership of our submission. If there are additional questions or concerns of clarity, please do not hesitate to contact our team immediately. As it was suggested we performed English editing

The manuscript entitled “Polymorphism within one of IGFBPs genes influence on pH values after 24h storage in different rabbit (Oryctolagus cuniculus) breeds” is important in terms of providing valuable information. I have made some recommendations for improving the proposed paper.

  1. The manuscript contains some syntax errors and misspellings. Revise the text to improve readability. Linguistic revision is strongly suggested. 

It was done

  1. When the article is examined, the aim of the article is stated by the authors as "Therefore, the current study analyze IGFBPs genes for possible polymorphisms associated with growth, slaughter, carcass and meat quality traits in different rabbits breed". The article title is "Polymorphism within one of IGFBPs genes influence on pH values after 24h storage in different rabbit (Oryctolagus cuniculus) breeds". This title is not suitable for the content and purpose of the article. The article title should be rearranged in accordance with the content and purpose of the article.

We suggest title “Polymorphism within  IGFBPs genes influence on the acidity, colour and shear force of rabbit meat”

  1. When Table 2 is examined, two different expressions (“-“ or “0”) are used to express the absence of some alleles and genotypes. One of these may be preferred here.

It was corrected – now only “-“ sign is present in table 2

  1. The conclusion section is written only for pH 24 and is quite insufficient in terms of comprehensiveness. However, in the purpose and content of the article, growth, slaughter, carcass, and meat quality characteristics were also examined. Therefore, the result needs to be expanded and rearranged to include all examined parameters.

We discussed results for other identified traits

  1. Abbreviations had been made, but these abbreviations were not explained in the article (for example Cins).

It was corrected – instead of Cins we added G/GAC as is should be properly named as a insertion

  1. The article provides very valuable information, but there are some problems in the presentation of this information. Especially the tables are given very confusingly. Understanding these tables is quite problematic. For this reason, redesigning the tables in a more understandable way will add value to the article.

we reorganized tables – they are on one page and we  divide breeds and we hope it will make easier to understand our results

Round 2

Reviewer 3 Report

Comments and Suggestions for Authors

The manuscript is corrected according suggestions and there are no further suggestions.

Reviewer 4 Report

Comments and Suggestions for Authors

It can be seen that the authors made all corrections requested by the referees in the article. In this case, the manuscript can be published in your journal. However, I have some advice for writers. There are still some grammar and spelling errors in the article (for example; in line 304, "rowandbedswith", line 429 "alack"). Before the article is published, they should review it and correct these errors.

Comments on the Quality of English Language

I have some advice for writers. There are still some grammar and spelling errors in the article (for example; in line 304, "rowandbedswith", line 429 "alack"). Before the article is published, they should review it and correct these errors.